# Understanding Kinesiophobia: Predictors and Influence on Early Functional Outcomes in Patients with Total Knee Arthroplasty

**DOI:** 10.3390/geriatrics9040103

**Published:** 2024-08-13

**Authors:** Milica Aleksić, Ivan Selaković, Sanja Tomanović Vujadinović, Marko Kadija, Darko Milovanović, Winfried Meissner, Ruth Zaslansky, Svetlana Srećković, Emilija Dubljanin-Raspopović

**Affiliations:** 1Center for Physical Medicine and Rehabilitation, University Clinical Center Serbia, 11000 Belgrade, Serbia; ivan.selakovic@gmail.com (I.S.); drsanjatv@gmail.com (S.T.V.); edubljaninraspopovic@gmail.com (E.D.-R.); 2Faculty of Medicine, University of Belgrade, 11000 Belgrade, Serbia; kadija.marko@gmail.com (M.K.); darkomil@doctor.com (D.M.); svetlanasreckovic@yahoo.com (S.S.); 3Clinic for Orthopedic Surgery and Traumatology, Clinical Center Serbia, 11000 Belgrade, Serbia; 4Department of Anesthesiology and Intensive Care, Jena University Hospital, Friedrich Schiller University, 07737 Jena, Germany; winfried.meissner@med.uni-jena.de (W.M.); ruth.zaslansky@gmail.com (R.Z.); 5Center for Anesthesiology and Resuscitation, University Clinical Center of Serbia, 11000 Belgrade, Serbia

**Keywords:** kinesiophobia, TKA, pain, functional recovery

## Abstract

This observational study aimed to identify predictors of kinesiophobia and examine its correlation with early functional outcomes in TKA recipients. On the first and fifth postoperative days (POD1 and POD5), we evaluated pain using the International Pain Outcomes Questionnaire (IPO-Q) and created multidimensional pain composite scores (PCSs). The Total Pain Composite Score (PCStotal) assesses the overall impact of pain, taking into account outcomes of pain intensity, pain-related interference with function, and emotions and side effects. Functional status on POD 5 was determined by the Barthel index, 6 min walking test, and knee range of motion. Kinesiophobia was assessed on POD5 using the Tampa Scale for Kinesiophobia (TSK). Among 75 TKA patients, 27% exhibited kinesiophobia. The final regression model highlighted PCStotal on POD5 (OR = 6.2, CI = 1.9–19.9), PCStotal (OR = 2.1, CI = 1.2–3.8) on POD1, and the intensity of chronic pain before surgery (OR = 1.4, CI = 1.1–2.1) as significant kinesiophobia predictors. On POD5, those with kinesiophobia showed increased dependency, slower gait, and poorer knee extension recovery. This study emphasizes the need to identify and address kinesiophobia in TKA patients for better functional outcomes and recovery. Additionally, it is vital to assess different domains of pain, not just pain intensity, as it can lead to kinesiophobia development.

## 1. Introduction

Total knee arthroplasty (TKA) is an effective treatment option for patients with end-stage knee osteoarthritis, providing pain relief, functional improvement, and health-related quality of life. Due to increasing levels of obesity, population aging, and growth in sports-related injuries, the incidence of TKA is high [1]. Between 2000 and 2014, the estimated annual numbers of primary TKA increased by 148% in the United States. Based on this data, Sloan et al. showed that the projected growth for TKA procedures will reach 935,000 annually by 2030 in the US [2].

The primary outcomes of the procedure are to improve the patient’s mobility in the postoperative period, enhance their functionality, alleviate pain significantly, and enhance self-confidence due to improved functionality [3].

However, TKA is a painful procedure. The number of patients reporting moderate to severe pain after TKA remained relatively constant, with 58% reporting moderate to severe pain on postoperative day 1 (POD1), which only decreased to 43% by postoperative day 3 (POD3) [4]. Pain in the postoperative period affects rehabilitation and increases the risk of complications in the acute phases and of developing chronic pain after surgery. Furthermore, up to 20% of patients are dissatisfied with the results of their surgery due to persistent pain and disability [5].

Additional psychological factors, such as pain catastrophizing, pain-related fear of movement, and depression, have been identified as contributors to prolonged pain and disability in individuals with different musculoskeletal conditions [6]. Patients may delay arthroplasty surgery due to the fear of acute postoperative pain [7]. Pain catastrophizing is a term used to describe the tendency to magnify the threat value of pain stimuli and to feel helpless in the context of pain. This can lead to an inability to inhibit pain-related thoughts during or after a painful encounter [8]. Kinesiophobia, on the other hand, is an excessive and debilitating fear of physical movement and activity due to a feeling of vulnerability from a painful injury or re-injury [9]. It is gaining more attention since it can lead to illness behavior and create a vicious cycle of pain and disability [10]. High levels of kinesiophobia after TKA negatively affect short- [11,12,13] and long-term [6,10,13,14,15,16,17] functional outcomes. Investigating the presence of kinesiophobia early after surgery can help arrange personalized treatment for this vulnerable group of patients.

Despite the growing interest in the relationship between kinesiophobia and TKA, there are limited studies on the etiology and psychological pattern of kinesiophobia in the literature. To the best of our knowledge, there are only two studies that examine the risk factors for the onset of kinesiophobia following TKA. Cai et al. showed that female sex, older age, lower levels of education, negative coping styles, lower self-efficacy, and pain were predictors of kinesiophobia after surgery [18]. Degirmenci et al. demonstrated that the choice of anesthesia techniques during total knee arthroplasty (TKA) significantly influences the development of postoperative kinesiophobia [19]. This study found that patients who received regional anesthesia and deep sedation were able to recover and move more confidently during the early postoperative period, while those who received regional anesthesia and light sedation experienced anxiety and fear, which made them hesitant to move [19]. Studies have highlighted the importance of kinesiophobia as a risk factor for higher pain intensity following TKA [10,11,15,16,17]. However, due to the lack of standardized pain measurements and the predominant use of unidimensional pain analysis, the relationship between patient-reported outcomes (PROs) and kinesiophobia has not been thoroughly investigated in any single study to date. PROs are reports coming directly from a patient about how they feel or function about a health condition and its therapy without interpretation by healthcare professionals or anyone else. PROs can relate to symptoms, signs, functional status, perceptions, or other aspects such as convenience and tolerability. PROs are not only important when more objective measures of disease outcome are not available but also represent what is most important to patients about a condition and its treatment [20]. Gewandter et al. suggested that the inclusion of multiple domains in the outcomes can be a significant advantage as it provides a more thorough evaluation of the experiences of the individuals under study, rather than relying on a single factor that may not be sufficient in describing their overall experience [21].

This study aimed to investigate the factors associated with kinesiophobia following TKA and to examine the relationship between kinesiophobia and early functional outcomes in TKA patients.

## 2. Materials and Methods

### 2.1. Setting

This observational study was conducted at the Clinic for Orthopedic Surgery and Traumatology, University Clinical Center Serbia in Belgrade over a period of 6 months. This study followed the principles of the Helsinki Declaration and was approved by the local ethics committee (Number 2017-004244-37). The findings in this study are based on the methodology outlined by PAIN OUT (www.pain-out.eu), which provides a user-friendly online system for hospitals to gather standardized patient feedback and clinical data using a questionnaire available in more than 20 languages. The registry offers tools to evaluate pain-related patient-reported outcomes (PROs) and management on the first day after surgery (POD1), as described in clinicaltrials.gov NCT02083835 [22,23]. Our research team was involved in creating the database used in this study, and this resource is also widely utilized by many hospitals worldwide.

Patients who had undergone TKA and who were 18 years or older, could communicate, and provided written consent were invited to participate in this study. The exclusion criteria included venous thromboembolism, neurological and musculoskeletal diseases that could affect recovery, and mental disorders. Written consent explained that the study aimed to improve pain treatment for patients after TKA in the future, and confirmed that no changes were being made to the standard medical care at the moment.

### 2.2. Surgical Technique, Anesthesia, Pain Management, and Postoperative Rehabilitation Program

The surgical procedure for TKA involved the insertion of tricompartmental prostheses using a standard medial parapatellar approach, with the use of cruciate-substituting designs. A femoral tourniquet at 300 mmHg was employed to achieve a bloodless surgical field. A compression bandage was applied from the toes to the mid-thigh at the end of the surgery. Spinal anesthesia with 10–15 mg levobupivacaine 0.5% or general anesthesia with propofol and fentanyl was administered during the procedure. Local infiltration anesthesia was not used in our study group. The regular protocol for pain management involved the scheduled assessment of pain and administration of non-opioid drugs (such as Paracetamol, Ketorolac, and Metamizol) and weak opioids (Tramadol) based on the severity of the pain reported by patients, and following the WHO’s approach to the use of analgesics based on pain severity. The pain was assessed at least once per shift. This treatment approach was implemented from POD1 to POD5.

All patients followed a standardized postoperative rehabilitation program beginning on POD1. Assisted ambulation and regular exercise to restore strength and mobility in the operated knee were performed 2 times a day for 20–30 min.

### 2.3. Data Collection

Patients were evaluated on POD1 and POD5.

(1)Baseline characteristics

On POD1, patients were assessed regarding demographic and clinical data comprising gender, year of birth (age), weight and height, intensity, and location of chronic pain before surgery. Furthermore, the type of anesthesia and duration of surgery were recorded.

(2)EuroQol-5D

Health-related quality of life during the last week before TKA was rated with the use of the EuroQol-5D (EQ5D) index score on POD1. After the surgery, we evaluated the patients’ overall well-being on POD5 using the same tool to determine the impact of TKA on their quality of life. An EQ5D index score of 0 indicates the worst possible health state and a value of 1 indicates full health [24].

(3)Multi-dimensional assessment of pain on POD1 and POD5

The validated International Pain Outcomes Questionnaire (IPO-Q) was used to evaluate pain-related PROs [23]. This questionnaire evaluates the following domains: intensity of pain and relief from treatments; interference of pain with physical activities in and out of bed; negative affect due to pain (anxiety and helplessness); adverse effects (AEs) (nausea, fatigue, dizziness, and itching); and perception of care (wish for more pain treatment, satisfaction with pain treatment, participation in decisions about pain treatment, and receipt of information about treatment). Pain intensity and pain-related physical and affective interference were quantified by patients using an 11-point numerical rating scale (0 = null, 10 = worst possible). The patient’s perception of care was assessed with yes or no on percentage scales. The data were collected by surveyors who underwent training before they approached patients. To reduce interviewer bias, patients completed the questionnaire independently with no assistance from family or staff. However, if a patient requested help, the surveyor could assist.

(4)Kinesiophobia on POD5

On POD5, kinesiophobia was measured with the Tampa Scale for Kinesiophobia (TSK). The TSK is a 17-item questionnaire designed to assess a patient’s fear of movement or (re)injury [25]. Each point has a 4-point Likert scale, scoring alternatives from “strongly disagree” to “strongly agree”. The total score on the TSK ranges from 17 to 68 [26]. We used a pre-validated cut-off score of 37 on the TSK to categorize knee replacement patients into two groups: those with no or low degrees of kinesiophobia (TSK‹37) and those with a high degree of kinesiophobia (TSK ≥ 37) [25,27].

(5)Functional outcome measures on POD5

On POD5, a functional assessment was conducted, which included three tests: knee range of motion (ROM), Barthel Index, and the 6 min walking test (6-MWT). The Barthel Index is an ordinal scale used to assess a person’s ability to perform ADL. It involves scoring 10 variables related to mobility and ADL, with a higher score indicating greater independence [28]. The 6-MWT measures functional walking capacity. During the test, the patients were asked to walk for 6 min, and the distance covered in meters was recorded [29]. Knee ROM was assessed using a universal goniometer, and the average peak knee flexion and extension were recorded from three trials. Health-related quality of life after TKA was rated using the EQ-5D index score.

### 2.4. Study Outcomes

The primary focus of this study was to identify predictors of kinesiophobia, while the secondary objective was to examine the association between kinesiophobia and functional outcomes.

### 2.5. Data Analysis

#### 2.5.1. Creating Multidimensional Composite Scores

Multidimensional composite scores were created based on ratings obtained from the IPO-Q. For POD1 and POD5, continuous PROs were extracted from the questionnaires and combined to form composite scores, as described by Hofer D et al. [30]. Three subscores were generated to assess pain intensity, pain-related interference, and side effects. The Pain Composite Score (PCS) was calculated using the formula: worst pain ×(% time in severe pain × 100) + least pain × (1 − % time in severe pain/100). The Pain Interference Total Score (PITS) was calculated as the mean of pain-related interference with activities in bed, breathing deeply/coughing, sleep, and pain-related anxiety and helplessness. The Pain Side Effects (PSE) composite score was calculated from the scores for dizziness, drowsiness, nausea, and itching [30]. The Total Pain Composite Score (PCStotal) was formulated by averaging the continuous items derived from the pain intensity, pain interference, and side effects domains of the IPO-Q [31].

#### 2.5.2. Statistics

Statistical analysis was conducted using the Statistical Package for the Social Sciences (SPSS Inc., Chicago, IL, USA) version 22.0. Data were visually analyzed with histograms, Q–Q plots, and Kolmogorov–Smirnov tests for normality of distribution. Categorical and dichotomous data were presented as absolute frequencies and percentage of patients. Continuous data were presented as the median, first quartile (Q1), and third quartile (Q3), and NRS scores as a median with the interquartile range. The chi-square test was applied to test relationships between categorical variables. A two-sided independent samples *t*-test was used to compare the mean values of normally distributed data between 2 groups. Ordinal data were compared by the 2-sided Mann–Whitney U test.

Univariable and multivariable logistic regression analyses were performed to assess the factors associated with kinesiophobia. Variables with a *p*-value <  0.20 in the unavailable analysis were retained and included in the multivariable regression, for which the backward selection method was used. Setting a threshold of 0.20 in the univariate analysis acts as an initial filter to capture a broader set of variables. This is crucial because some variables might show weak individual associations but can become significant when adjusted for other variables in a multivariable model [32,33,34]. In the backward method, the model started with all variables in the equation. Using criteria for removal, variables that did not contribute to the solution were removed one at a time. The variable with the smallest partial correlation was taken out first. The steps proceed until no remaining variables are qualified for removal [35].

No collinearity problem was detected for any of the models. In all instances, a *p*-value < 0.05 was considered statistically significant.

## 3. Results

### 3.1. Baseline Characteristics of the Study Group

A total of 81 patients were recruited for the study. During the study period, three patients declined to fill out the requested questionnaires, and three patients were excluded due to deep vein thrombosis. Therefore, 75 patients were included in the study analysis.

The patients were categorized into two groups based on their degree of kinesiophobia: a high kinesiophobia group (*n* = 20) and a low kinesiophobia group (*n* = 55).

Patient characteristics and clinical data are summarized in Table 1.

### 3.2. Analysis of Group Differences Regarding Pain-Related Outcomes and Health-Related Quality of Life Outcomes

A significant difference in PROs at POD1 and POD5 was observed. Patients with kinesiophobia demonstrated significantly worse pain outcomes on the PCStotal of IPO-Q compared to patients without kinesiophobia on POD1 and POD5. We have provided data on the individual PROs as well. According to our findings, individuals with kinesiophobia exhibit higher scores on PCS and PITS subscales on POD1 and POD5, whereas no significant difference was observed in the results of PSE (Table 2).

Patients with kinesiophobia also had a lower percentage of pain relief, wished for more analgesics, and reported more interference of pain with activities out of bed on POD5.

A statistically significant difference in EQ-5D on POD5 between the groups was also observed (Table 3).

### 3.3. Predictors of Kinesiophobia

Univariable regression analysis revealed that patients who reported higher pain intensity before surgery and had worse quality of life preoperatively were more likely to develop kinesiophobia. As far as pain-related PROs are concerned, our results revealed that higher scores of PCStotal of IPO-Q were associated with higher kinesiophobia scores. Furthermore, a lower percentage of pain relief, wish for more pain treatment, interference of pain with activities out of bed, and higher scores on EQ-5D on POD5 were also related to higher kinesiophobia scores (Table 4).

Table 5 displays the results of the multivariable regression analysis. The final model included intensity of chronic pain before surgery, PCStotal on POD1 and POD5, and pain interfering with activities out of bed as significant predictors. With the independent variables added, the overall model was statistically significant (χ^2^ = 28.286, *p* < 0.001). The model explained 45.8% (Nagelkerkes R^2^) of the variance of kinesiophobia and correctly classified 80% of cases. The strongest predictor of kinesiophobia was PCStotal on POD 5, whose odds ratio (OR) was 6.191 when adjusted for PCStotal on POD1, intensity of chronic pain before surgery, and pain interfering with activities out of bed. PCStotal on POD1 was identified as the second strongest predictor of kinesiophobia.

### 3.4. Influence of Kinesiophobia on Recovery after TKA

Regarding functional outcomes on POD5, patients with kinesiophobia revealed significantly higher dependency levels as expressed with the Barthel score, had a slower gait speed on the 6-MWT, and showed worse recovery of knee extension (Table 6).

According to the results of univariable regression analysis, a significant association was found between the presence of kinesiophobia in patients and a slower rate of recovery (Table 7).

## 4. Discussion

In this study, we observed a kinesiophobia incidence of 27%, which is close to the rates reported in earlier studies in TKA patients in Serbia (22%) [16] and China (24%) [18]. Our findings indicate that the presence of kinesiophobia might be impacted by the intensity of preoperative pain. This observation is consistent with Kroska’s et al.’s postulation that fear avoidance behavior is often associated with higher pain intensity [36].

The most important finding of our study was the confirmed link between PCStotal on POD1 and POD5 and the development of kinesiophobia. To the best of our knowledge, our results are the first to highlight the relationship of both the intensity of postoperative pain and the physical and emotional interference caused by pain after TKA, as they are associated with the presence of kinesiophobia. Prior studies on TKA patients primarily relied on pain intensity and used single-dimensional measures to assess pain [16,18,19]. Moreover, research on risk factors for kinesiophobia after TKA found a direct link between high pain intensity levels within the first 24 h after surgery and increased levels of kinesiophobia [18]. Composite scores for pain, which combine pain intensity, pain-related interference, and side effects, offer a unique approach that provides a holistic view of the pain experience [21,30]. This approach provides a broader perspective compared to using a single measure [21].

To evaluate the influence of kinesiophobia on early functional outcomes, we used the 6-MTW, knee ROM, and Barthel index. Our study’s findings support previous research regarding the 6-MWT and its relation to kinesiophobia. Doury-Panchout et al. demonstrated that patients without kinesiophobia walked a significantly greater distance during the 6-MWT compared to those with kinesiophobia [15]. Additionally, Guney Deniz et al. found a positive correlation between higher TSK scores and improved 2-MWT scores [12]. Similarly, Degirmenci et al. discovered that higher TSK scores were associated with better 2-MWT scores and Timed Up and Go (TUG) test results on POD 2 and POD 5 [19]. Based on our findings, it appears that there might be an inverse correlation between active knee extension and kinesiophobia, while no correlation was observed with knee flexion. However, it is important to note that not all studies align with these results. Active knee flexion was found to be correlated with TSK in several studies [12,16,19,37]. In contrast, Doury-Panchout et al. did not observe any notable disparity in maximum passive flexion and maximum active extension on the day of discharge between high-TSK and low-TSK groups [15]. Similarly, Filardo et al. found no connection between high-TSK and low-TSK groups concerning active or passive ROM [38]. In addition, our study revealed the negative relationship between higher levels of kinesiophobia and functional independence as measured with the Barthel index on POD5.

## 5. Study Strengths and Limitations

The main strength of our study is our innovative methodological approach. To the best of our knowledge, this is the first study to evaluate the multidimensional impact of pain on kinesiophobia. There are several limitations of our study. First, the composite scores used in our study require calculations and are therefore not appropriate as a tool in everyday clinical routine. Second, the sample size is relatively small. Third, the participants in this study are exclusively from a single hospital in Serbia. Therefore, it remains uncertain how these findings can be extrapolated to a wider general population. Finally, the pre-surgery kinesiophobia scores were not assessed.

## 6. Conclusions

Our research findings reveal a high prevalence of kinesiophobia after TKA, highlighting its importance during postoperative care. Moreover, our study identifies pain as a significant predictor of kinesiophobia and highlIghts its impact on poor functional outcomes after surgery. Notably, composite scores for pain evaluation prove to be superior to unidimensional scales, offering a more comprehensive approach to understanding the connection between pain and kinesiophobia. By identifying individuals prone to kinesiophobia through multidimensional pain assessment, healthcare professionals can adjust strategies to improve outcomes and post-surgery recovery. Further research is expected to show the influence of improved pain treatment strategies on kinesiophobia levels in patients after TKA and to quantify the impact of individual PROs on kinesiophobia.

## Figures and Tables

**Table 1 geriatrics-09-00103-t001:** Sample description for Continuous and Dichotomous Variables.

Variable	Mean	SD	
Age ^1^	67.61	±8.102	
Variable	median	Q1	Q3
Age (y.) ^2^	68	63	74
Weight (kg) ^2^	82	70	90
BMI (kg/m^2^) ^2^	28.4	25.3	32
Duration of surgery ^2^	138	120	165
Intensity of chronic pain before admission ^2^	6	5	8
EQ5D preoperative ^2,a^	0.636	0.416	0.750
Total Pain Composite Score (PCStotal) POD1 ^2^	2.07	1.31	2.79
Total Pain Composite Score (PCStotal) POD5 ^2^	0.41	0.09	1.11
Variable	N	%	-
Gender ^3^			
Male	20	27	
Female	55	73	
Marital status ^3^			
Married	69	92	
Single	6	8	
Education ^3^			
Primary school or under	18	24	
Secondary school	44	59	
College	13	17	
Type of anesthesia ^3^			
General	29	39	
Spinal	46	61	
Nonopioid administered (ward) ^3^		96	
Paracetamol	71	48	
Ketoprofen	36	20	
Ketrolac	15	43	
Metamizole	32	8	
	6		
Systemic opioid (ward) ^3^	63	85	

^1^ = the values are given as the mean ± standard deviation (SD); ^2^ = the values are given as numerical rating scale scores by median with interquartile range (IQR); ^3^ = The values are given as the number of patients with the percentage in parentheses. ^a^: EQ5D = EuroQol-5D.

**Table 2 geriatrics-09-00103-t002:** Comparasion of total composite scores and subscores between TKA patients with and without kinesiophobia.

Variable ^1^	Low Kinesiophobia N (55)			High Kinesiophobia (20)			*p*-Value *
	Median	Q1	Q3	Median	Q1	Q3	
POD1							
PCStotal 1 ^a^	1.59	1.10	2.52	2.76	2.33	3.71	0.002
PCS 1 ^b^	2.6	1.40	4.00	4.7	3.35	5.55	0.002
PITS1 ^c^	1	1.00	2.00	2.00	1.00	3.75	0.039
PSE1 ^d^	0.50	0.00	2.00	1.12	0.00	2.75	0.180
POD5							
PCStotal2 ^e^	0.31	0.09	0.55	1.28	0.59	2.15	0.000
PCS2 ^f^	0.00	0.00	0.70	2.35	0.90	3.60	0.000
PITS2 ^g^	0.43	0.14	0.43	1.57	1.00	2.14	0.000
PSE2 ^h^	0.00	0.00	0.00	0.00	0.00	0.50	0.166

^1^ = The values are given as numerical rating scale scores by a median with interquartile range (IQR); * Mann–Whitney U test; ^a^: PCStotal 1 = composite score of patient-reported outcome measures on POD1; ^b^: PCStotal 1 = the Pain Composite Score on POD1; ^c^: PITS1 = the Pain Interference Total Score on POD1; ^d^: PSE1 = the Pain Side Effects composite score on POD1; ^e^: PCStotal2 = composite score of patient-reported outcome measures on POD5; ^f^: PCS2 = the Pain Composite Score on POD5; ^g^: PITS2 = the Pain Interference Total Score on POD5; ^h^: PSE2 = the Pain Side Effects composite score on POD5.

**Table 3 geriatrics-09-00103-t003:** Comparison of PROs that are not included in composite scores between TKA patients with or without kinesiophobia.

Variable	Low Kinesiophobia N (55)			High Kinesiophobia (20)			*p*-Value *
	Median	Q1	Q3	Median	Q1	Q3	
Outcomes that are not included in composite scores on POD1							
Pain interfering with activities out of bed ^1^	3	2	5	5	3.5	6.5	0.811
Participation in decisions regarding pain treatment ^1^	9	6	10	7.5	6	10	0.355
Satisfied with the result of pain treatment ^1^	9	8	10	9	7	9	0.108
	mean	SD		mean	SD		
Percentage of pain relief ^2^	75.82	±21.40	-	69.00	±25.11	-	0.205
	N	%		N	%		
Desire more pain treatment ^3^							
Yes	29	53		11	55		1.000 **
No	26	47		9	45		
Percentage of patients getting out of bed ^3^							
Yes	34	62		13	65		1.000 **
No	21	38		7	35		
Outcomes that are not included in composite scores on POD5							
Pain interfering with activities out of bed ^1^	1	0	3	3	1	5	0.047
Participation in decisions regarding pain treatment ^1^	9	9	10	9	4.25	10	0.518
Satisfied with the result of pain treatment ^1^	9	9	10	9.5	8.25	10	0.644
	mean	SD		mean	SD		
Percentage of pain relief ^2^	86.48	±17.82	-	68.75	±18.21	-	0.001
	N	%		N	%		
Desire more pain treatment ^3^							
Yes	3	5		6	30		0.003 **
No	52	95		14	70		
Health-related quality of life							
	mean	SD		mean	SD		
EQ5D preoperative ^a^	0.599	±0.209		0.467	±0.267		0.052
EQ5D POD5	0.674	±0.141		0.539	±0.218		0.000

^1^ = the values are given as numerical rating scale scores by median with interquartile range (IQR); ^2^ = the values are given as the mean ± standard deviation (SD); ^3^ = The values are given as the number of patients with percentages; * Mann–Whitney U test; ** chi-square test; ^a^: EQ5D = EuroQol-5D.

**Table 4 geriatrics-09-00103-t004:** Univariable prediction model of kinesiophobia.

	OR	95% CI	*p*-Value
EQ5D preoperative ^a^	0.087	0.009–0.860	0.037
Intensity of chronic pain before surgery	1.359	1.016–1.817	0.038
PCStoal 1 ^b^	2.183	1.289–3.696	0.004
PCStotal 2 ^c^	3.051	1.546–6.022	0.001
Percentage of pain relief on POD5	0.974	0.950–0.998	0.037
Pain interfering with activities out of bed POD5	1.272	1.015–1.549	0.037
Desire more pain treatment on POD5	0.107	0.024–0.472	0.003
EQ5D POD5	0.014	0.001–0.378	0.011

^a^: EQ5D = EuroQol-5D; ^b^: PCStotal 1 = composite score of patient-reported outcome measure on POD1; ^c^: PCStotal 2 = composite score of patient-reported outcome measures on POD5; OR: Odds ratio; 95% CI confidence interval.

**Table 5 geriatrics-09-00103-t005:** Multivariable prediction model of kinesiophobia.

	OR	95% CI	*p*-Value
PCStotal 1 ^a^	2.139	1.202–3.807	0.010
PCStotal 2 ^b^	6.191	1.918–19.979	0.002
Intensity of chronic pain before surgery	1.462	1.017–2.103	0.040
Pain interfering with activities out of bed	0.692	0.444–1.080	0.105

^a^: PCStotal 1 = composite score of patient-reported outcome measure on POD1; ^b^: PCStotal 2 = composite score of patient-reported outcome measures on POD5; OR: Odds ratio; 95% CI confidence interval.

**Table 6 geriatrics-09-00103-t006:** Difference between groups regarding functional outcome.

Variable	Without Kinesiophobia N (55)		With Kinesiophobia (20)		*p*-Value ^1^
	Mean	SD	Mean	SD	
Barthel	77	11.21	66.25	13.66	0.001
6-MTW ^a^	112.64	45.51	74.90	50.35	0.002
Extension	−16.64	9.33	−22.25	7.86	0.020
Flexion	65.55	17.73	63.25	14.26	0.468

^a^: 6-MWT = 6 min walking test; ^1^ = two-sided independent sample *t*-test.

**Table 7 geriatrics-09-00103-t007:** Univariable prediction model of functional recovery.

	B	95% CI	*p*-Value
6-MWT ^a^	−0.340	−32.102; −13.371	0.000
BARTHEL	−0.376	−16.939; −4.561	0.000
EXTENSION	−0.270	−10.283; −0.944	0.019
FLEXION	−0.061	−11.090; 6.499	0.604

^a^: 6-MWT = 6 min walking test.

## Data Availability

The raw data supporting the conclusions of this article will be made available by the authors on request.

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
