# Peer review of "Understanding Kinesiophobia: Predictors and Influence on Early Functional Outcomes in Patients with Total Knee Arthroplasty"

_geriatrics, 2024, doi:10.3390/geriatrics9040103_

Round 1

Reviewer 1 Report

Comments and Suggestions for Authors

Dear authors, all comments are in the attached file.

Comments on the Quality of English Language

Author Response

Comments 1: Please expand on the topic regarding the website www.pain-out.eu included in the methodology. Did you use the data on this website or did you contribute to the creation of the databases on this website? Are you the only one who relied on the methods contained on this website?

Response 1: According to the reviewer's suggestion, we have included a detailed paragraph in the methodology section (section 2.1) explaining the Pain-Out website and our contribution to creating the databases.

   ,,The findings in this study are based on the methodology outlined by PAIN OUT (www.pain-out.eu), which provides a user-friendly online system for hospitals to gather standardized patient feedback and clinical data using a questionnaire available in more than 20 languages. The registry offers tools to evaluate pain-related patient-reported outcomes (PROs) and management on the first day after surgery (POD1), as described in clinicaltrials.gov NCT02083835 [22,23]. Our research team was involved in creating the database used in this study, and this resource is also widely utilized by many hospitals worldwide.

Comments 2: Please explain why variables that would have a p-value<0.20 in a univariate regression analysis were included in the multiple regression analysis?

Response 2: Thank you for this valuable comment.

Setting a higher threshold of 0.20 in the univariate analysis acts as an initial filter to capture a broader set of variables. This is crucial because some variables might show weak individual associations but can become significant when adjusted for other variables in a multivariable model​. This explanation with the related references was added in the statistics (2.5.2) section of the manuscript.

Comments 3: Please provide the exact age of the patients, because the methodology contains information that these were patients aged 18 and over, and in the table the median is 68, Q1-63, Q3-74 - this should be unified in both places. It is also suggested to replace the median and quartile range values with the mean and standard deviation.

Response 3: We stated that patients aged 18 and over are allowed to participate in the study (inclusion criteria). However, knee OA is predominantly seen in older patients. This is why the mean age in our group 67.61 years (SD ±8.102). We changed the values in the table 1 from median to mean values.

Comments 4: The "marks" in the tables should be corrected, e.g. $ etc., because they are confusing for the reader. If you want to explain something under the table, please use e.g. the letters: a, b, c,… or the numbers 1, 2, 3,… in the superscript.

Response 4: Thank you for pointing this out. We have corrected the notations in all tables, using letters (a, b, c,…) or numbers (1, 2, 3,…) in the superscript to provide explanations.

Comments 5: For p-value in tests comparing quantitative values, please indicate in the upper index which test was used - Student's t-test or Mann-Whitney U.

Response 5: The specific tests used have been indicated in the tables.

Comments 6: Under each table, the abbreviations contained therein should be explained; currently, their explanations must be found in the text of the manuscript, which is cumbersome for the reader understanding the article.

Response 6:  We agree with this comment. Therefore, we have added explanations for all abbreviations directly under each table.

Comments 7: The number and size of tables "overwhelms" the reader a bit when reading the work, making it less transparent - please consider transferring e.g. characteristics or descriptive statistics to additional material, and leave only the basic dependencies in the work.

Response 7: Thank you for your comment. We realize that the material is overwhelming, but we were sticking to the journal's guidelines stating that all tables need to be enlaced in the main text. If there is place to change something please let us know.

Comments 8: The study group is rather small, which the authors themselves mention in the limitations of the study. Maybe it would be worth enlarging the group, because it is a survey study after all. Considering a survey, this group is absolutely too small to be published at this stage. I believe that the authors should increase the study group to at least 200 so that preliminary conclusions can be drawn.

Response 8:  Thank you for this valuable comment about the study group size. We acknowledge that the study group is relatively small, and we stated this in our study limitations. Our sample size was based on the availability and willingness of patients to participate during the study period. We highly value the suggestion to increase the study group size to at least 200 for more robust conclusions in the future in a new study, but we cannot change our study group and results now. Despite this limitation, we firmly believe that our findings provide valuable preliminary insights that can guide future research and clinical practice. Therefore, we would highly appreciate the possibility to publish our results.

Comments 9: The references should be corrected, standardized and prepared in accordance with the journal's recommendations.

Response 9: We have revised the references to ensure they are in accordance with the journal's guidelines.

Reviewer 2 Report

Comments and Suggestions for Authors

The present study identified and addressed the Kinesiophobia in patents undergone Total Knee Arthroplasty. The study was conducted well, and results are presented and discussed clearly in the manuscript.

I have few minor comments to authors before its acceptance for publications.

1.      Did authors compared Kinesiophobia at baseline compared to 6month follow-up in patients

2.      Is there specific reason for selecting older population for the study? In general, kinesiophobia is more in the older population.

3.      Why study was conducted in less population (75 only).

4.      It would be more impressive if the study would have been conducted in different regions of the country covering various demographics.

5.      Any of the study population had prior psychological problems like depression and anxiety? as these conditions impact the outcome.

6.      Inclusion and exclusion criteria for the study?

Author Response

Comments 1: Did authors compared Kinesiophobia at baseline compared to 6month follow-up in patients

Response 1: Thank you for this question. Comparing kinesiophobia at baseline and the 6-month follow-up was not the aim of our study. However, this can be a valuable suggestion for future research.

Comments 2: Is there specific reason for selecting older population for the study? In general, kinesiophobia is more in the older population.

Response 2: The participants in our study were selected because they had undergone total knee arthroplasty due to knee osteoarthritis (inclusion criteria), which is most common in the elderly population.

Comments 3: Why study was conducted in less population (75 only).

Response 3: We appreciate your feedback. We acknowledge that the study group is relatively small, and we stated this in our study limitations. Our sample size was based on the availability and willingness of patients to participate during the study period. Despite this limitation, we firmly believe that our findings provide valuable preliminary insights that can guide future research and clinical practice. Therefore, we would highly appreciate the possibility to publish our results.

Comments 4: It would be more impressive if the study would have been conducted in different regions of the country covering various demographics.

Response 4: Thank you for this valuable suggestion. We will definitely consider conducting future research in different regions to cover various demographics.

Comments 5: Any of the study population had prior psychological problems like depression and anxiety? as these conditions impact the outcome.

Response 5:  We did not examine prior psychological conditions like depression and anxiety in our cohort. However, we agree that investigating these conditions would be valuable for future studies.

Comment 6: Inclusion and exclusion criteria for the study?

Response 6:  Thank you for bringing this to our attention. We invited patients who were 18 years or older, had undergone total knee arthroplasty (TKA), could communicate effectively, and provided written consent to participate in the study. Exclusion criteria included venous thromboembolism, neurological and musculoskeletal diseases that could affect recovery, and mental disorders. Exclusion criteria are added to the manuscript (section 2.1) according to your suggestion.

Round 2

Reviewer 1 Report

Comments and Suggestions for Authors

Dear Authors,

Thank you for your responses and corrections. I will recommend your article for publication.

Your sincerelly,

Reviewer

Reviewer 2 Report

Comments and Suggestions for Authors

Thank you for providing the responses to reviewer's comments. The responses provided by the authors are satisfactory and I request editorial team to accept the manuscript in its present form.